# Key Factors Assessment on Bird Strike Density Distribution in Airport Habitats: Spatial Heterogeneity and Geographically Weighted Regression Model

**Quan Shao [1],\*, Yan Zhou [1], Pei Zhu [1], Yan Ma [2] and Mengxue Shao [1]**

[1]  College of Civil Aviation/College of Flight, Nanjing University of Aeronautics and Astronautics, Nanjing 210016, China; zhouyan_1127@163.com (Y.Z.); zpei@nuaa.edu.cn (P.Z.); 13101888133@163.com (M.S.)

[2]  College of Urban and Environmental Sciences, Peking University, Beijing 100871, China; maxyan@pku.edu.cn

\*  Correspondence: shaoquan@nuaa.edu.cn

**Abstract:** Although the factors influencing bird strikes have been studied extensively, few works focused on the spatial variations in bird strikes affected by factors due to the difference in the geographical environment around the airport. In this paper, the bird strike density distribution of different seasons affected by factors in a rectangular region of 800 square kilometers centered on the Xi'an Airport runway was investigated based on collected bird strike data. The ordinary least square (OLS) model was used to analyze the global effects of different factors, and the Geographically Weighted Regression (GWR) model was used to analyze the spatial variations in the factors of bird strike density. The results showed that key factors on the kernel density of bird strikes showed evident spatial heterogeneity and the seasonal difference in the different habitats. Based on the results of the study, airport managers are provided with some specific defense measures to reduce the number of bird strikes from the two aspects of expelling birds on the airfield area and reducing the attractiveness of habitats outside the airport to birds, so that achieve the sustainable and safe development of civil aviation and the ecological environment.

**Keywords:** bird strike density; spatial heterogeneity; key factors; airport habitats; GWR model

## 1. Introduction

Bird strikes are one of the challenges faced by airports around the world, which seriously affects the safe operation of airports [1–3]. The species, distribution, and ecological environment of birds in and around the airport are directly related to the number of bird strikes [4]. There are many kinds of researches on the control measures of bird strikes, such as the use of falconer [5], solar photovoltaic devices [6], and protective net [7] to reduce the number of birds around the airport and the possibility of bird strikes. In addition, since many birds' lives are based on mobility, each bird species occupies a niche within nature and its behavior varies with season, time of day, weather, and other factors [8]. For instance, urbanization as a major cause of biotic homogenization, and the "urban-adaptable" species become increasingly widespread and locally abundant in cities [9]. And the human population density affects vegetation coverage, which has an impact on bird species richness in turn [10]. Meanwhile, other researchers have proposed some studies on the relationship between airport habitats and bird diversity [11]. From this point of view, a single management framework combining two potentially conflicting objectives was proposed to promote conservation of obligate grassland birds and managing

to reduce bird hazards to aviation safety [12]. Other scholars investigated wildlife species (including bird populations) around the airport, and they found the species that most likely to collide with aircraft and proposed to increase patrols during the peak period (July–November) of the collision [13]. In light of the ecological environment of the airport, the attraction risk index has been ranked by Coccon et al. [14] and obtained that the attraction of different habitats to bird species was changed in a different season. In addition, the landscape structure in and around the airport has been proved to be one of the influencing factors leading to bird strikes [15].

Most of these studies that have analyzed the influencing factors of bird strikes depended on traditional regression models, failing to discover the spatial variations in these effects. Hence, they ignored the fact that the influence of factors may differ between different locations because of different environmental conditions and development states around the airport [16]. And the change of the relationship or structure between variables caused by the shift in geographical location is called spatial non-stationarity [17]. Then, as a local statistical method, Geographically Weighted Regression (GWR) model [18] explores the relationship between the independent and dependent variables in the spatial dimension by embedding the spatial position of the data into the regression parameters, which could solve the problem of spatial non-stationarity effectively [17]. And the GWR model has been widely used in other fields, such as transportation [19] and environment [20], and has achieved good results about the analytical ability and fitting effect of spatial data. Instead of creating a set of model parameters for the overall study object, the GWR model allows us to discover local variability in a statistical relationship, based on the hypothesis that closer observations have a greater impact on the parameters than farther comments [16]. Hence, in this paper, we will also use the GWR model to explore the spatial effects of the factors affecting bird strike density to fill the gap in the space research of bird strike safety.

Therefore, this paper aims to make a comprehensive spatial analysis of factors on bird strikes in the airport habitat. In this paper, the bird strike density as the dependent variable, and the factors within a rectangular region of 800 square kilometers centered on Xi'an Airport were selected as independent variables, such as population density, vegetation coverage, water distribution, and bird diversity. The spatial heterogeneity of key factors in the ecological environment of Xi'an Airport was studied based on the GWR model. The spatial distribution characteristics and the degree of effect each factor were obtained and analyzed. The results are expected to provide a reference for the management of the ecological environment around the airport to reduce the incidence of bird strikes and improve the safe development of civil aviation.

## 2. Materials and Methods

### 2.1. Study Area

In this paper, the spatial heterogeneity of bird strike density is analyzed by taking Xi'an Airport as an example. Xi'an Xianyang International Airport is located in North China, is one of the eight regional hub airports in China. The geographic features of the study area can be seen from Figure 1, which shows the study area is typical farmland and village landscape. Besides, the study area belongs to the warm temperate monsoon zone with a semi-humid climate and an average annual temperature of 14 °C. Based on the flight procedures, air route, and historical bird strikes of Xi'an Airport, the study area of this paper is defined as a rectangular area with a length of 40 km and a width of 20 km, to accommodate all the historical bird strikes occurred in the Xi'an Airport. Next, the data of bird strike points were obtained from the airport collected by pilots and professionals of damage prevention in 2017. In data processing, the study area was divided by a grid of 1 km × 1 km.

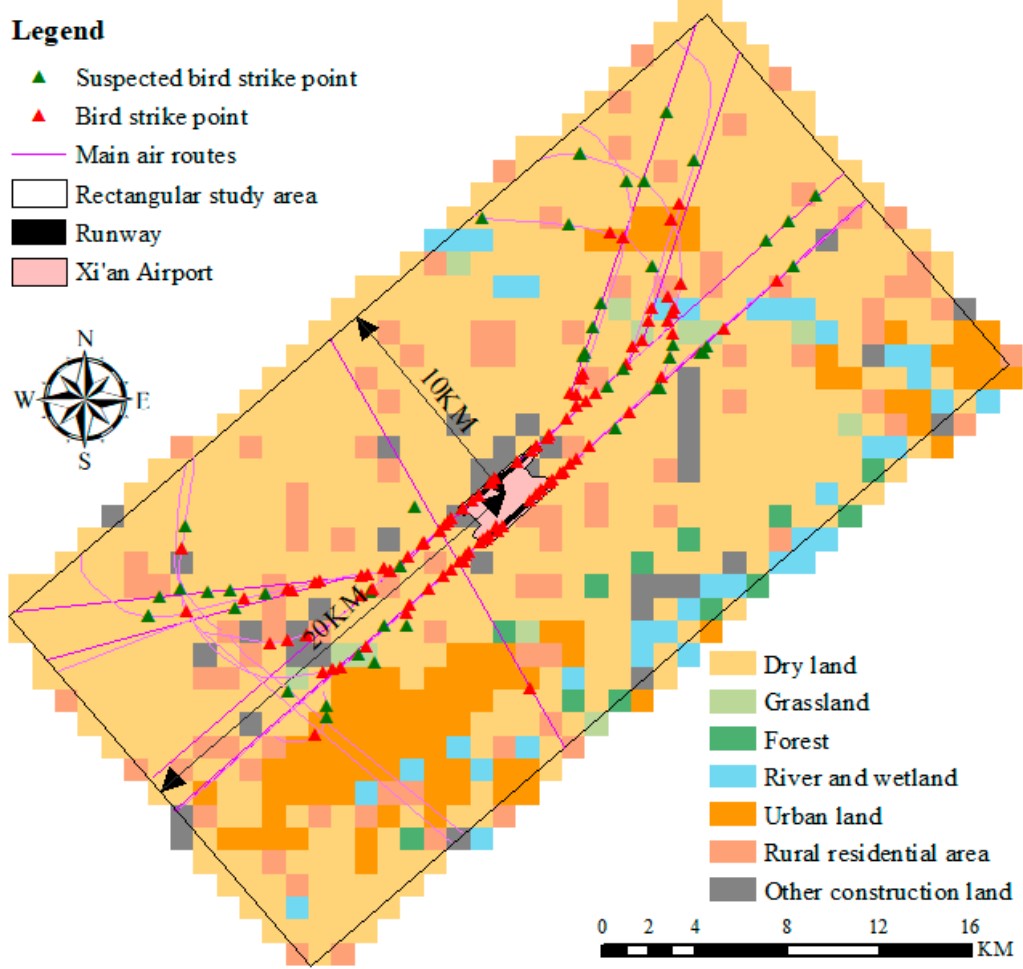

**Figure 1.** Land use and spatial distribution map of bird strike point at Xi'an Airport in 2017, which shows the area and orientation of the Xi'an Airport, two parallel runways operating independently, the main air route, the bird strike points, the river, the wetland, and the urban area, etc.

### 2.2. Dependent Variable and Independent Variables

The occurrence of bird strikes has seasonal characteristics [21], and Xi'an Airport is not an exception. Therefore, this paper will analyze the spatial heterogeneity of the dominant factors of bird strikes according to four seasons.

(1) Dependent variable: Kernel Density Estimation of bird strike. In order to facilitate the data analysis, this paper uses the Kernel Density Estimation (KDE) [22] to process the spatial distribution of bird strike points in each season, so that the discrete points in space become a continuous density surface, which not only takes into account the uncertainty of bird strike points and the error after positioning but also facilitates data sampling in the process of modeling and analysis. A total of 160 bird strikes were received at Xi'an Airport in 2017, of which 48 were suspected, and 112 were confirmed. Among them, the number of bird strikes in different seasons are 30 in spring, 42 in summer, 79 in autumn, and 9 in winter, respectively. Taking the autumn of 2017 as an example, the kernel density estimation statistics of bird strike points are shown in Table 1, and data for other seasons are in the supporting information (Table S1).

(2) Independent variables. According to the niche theory [23], birds that pass through, enter or inhabit the airport possess their spatial location and functional relationship with related biological populations, there are factors that attract birds to the airport environment (e.g., food, water and habitat). Then by summarizing numerous references and according to the characteristics of bird strikes [10,15,24–26], a total of 4 key environmental variables covering the majority of bird strike impact

factors are selected in this section, taking autumn as an example, the variable statistics are shown in Table 1.

**Table 1.** The descriptive statistics of variables in autumn (September-November).

| Variable | Description | Min | Median | Mean | Max | Std. Dev |
|---|---|---|---|---|---|---|
| KDE [1] | Kernel Density Estimation of bird strike (bird strikes/km$^2$) | 0.00 | 0.01 | 0.09 | 0.66 | 0.14 |
| P [2] | Human population density (people/km$^2$) | 466.6 | 1576.8 | 1463.0 | 4000.9 | 663.77 |
| NDVI [2] | Normalized difference vegetation index | 0.31 | 0.65 | 0.62 | 0.82 | 0.12 |
| BD [3] | Bird diversity index | 0.46 | 0.81 | 0.80 | 1.44 | 0.19 |
| RNE [1] | The reciprocal of the nearest Euclidean distance from the center of the grid to the water body (1/km) | 0.06 | 0.21 | 0.92 | 137.94 | 5.40 |

Notes: [1] ArcMap 10.5 Spatial Analysis Toolbox (See File S1); [2] Resource and Environment Data Cloud Platform (See File S1); [3] See File S2.

(a) Human population density: We chose the population density to represent human aggregation in the airport habitat [27], the average population density per square kilometer in a given grid, each season during the study period.

(b) Normalized Difference Vegetation Index (NDVI) variable: NDVI is recognized as a valid parameter to characterize vegetation change, and it contains useful information about vegetation cover [28,29]. The range of NDVI values is: $-1 \leq NDVI \leq 1$, negative values indicate that the ground is covered by clouds, water, snow, etc., 0 indicates the presence of rock or bare soil, etc., and a positive value indicates that there is vegetation cover and increases with the increase of vegetation coverage.

(c) Bird diversity index variable: The bird diversity index is calculated by Shannon index ($H'$), as Equation (1) [30].

$$H' = -\sum_{i=1}^{s} P_i \ln(P_i) \tag{1}$$

where $H'$ is the bird diversity index, $S$ is the number of bird species, $P_i$ is the proportion of the number of individuals of species $i$ to the number of individuals of all species. The bird diversity index is generally between 1.5 and 3.5, rarely more than 4.5 [31]. Xi'an Airport adopted leading the belt transect method and the fixed-radius sample point method to survey and collect the number and distribution of bird species. In this paper, the bird species observation data (S2 File) are used to calculate the bird diversity index with Equation (1). And the bird diversity index distribution in the study area is obtained by using the tools of "Kriging" and "Extract Multi Values to Points" in ArcMap 10.5. Addition

(d) The reciprocal of the nearest Euclidean distance between the center of the grid and the water area: This factor takes into account the influence of the spatial distance constraint of the water body on the region where the bird strike occurred. According to Tobler's First Law of Geography, the effect of water on other areas is defined by the reciprocal of the nearest Euclidean distance between the two, that is, the farther away the area is from the water, the less it is affected by the water. The nearest Euclidean distance from the center of the grid to the water body is obtained by the "Nearest Neighbor Analysis" tool in ArcMap 10.5, in kilometers.

## 2.3. Spatial Autocorrelation

Spatial autocorrelation refers to the potential interdependence between the observed data of some variables in the same distribution region [32]. Spatial autocorrelation analysis is used for the estimation and analysis of spatial dependence and heterogeneity among variables, commonly by means of the use of Moran's I index. Moran's I index is divided into global Moran's I index and local Moran's I index [20]. Among them, global Moran's I index describes the overall distribution of bird strike or other variables, while local Moran's I index reflects the aggregation differences between neighboring regions. Both two Moran's I index values are between −1 and 1. If Moran's I > 0, it means positive

spatial correlation, and the larger the value, the more obvious the spatial correlation. If Moran's I < 0, it indicates a negative spatial correlation, and the smaller the value, the greater the spatial difference, otherwise, Moran's I = 0, which means that the spatial distribution is random. Then, z-score and *p*-value represent the level of spatial correlation between neighboring regions (Figure S1) and their corresponding significant levels (Table S2), respectively. In addition, a High-High (HH) cluster type is composed of geographic units with a higher local mean (the average z-score of the target unit's neighbors) than the global mean (the average z-score of all geographic units studied); those geographic units which the local mean is lower than the global mean are classified as belonging to Low-Low (LL) cluster. When the z-score is higher than the local mean, the units are categorized as High-Low (HL) cluster, and the Low-High (LH) cluster type consists of units whose z-score is lower than the local mean [33,34].

### 2.4. The Geographically Weighted Regression Model

In the process of obtaining sample data, the observed data are generally sampled according to a given geographical location as a sampling unit, and the relationship or structure of variables will change with the change of geographical location. If we assume that the data of these variables are spatially uncorrelated, then the regression parameters of each variable can be obtained by the Ordinary Least Square (OLS), but the regression parameter value is the average value of the whole study area, which disguises the spatial variation characteristics of the regression parameters. In order to solve these problems in the classical regression model, foreign researchers have proposed the Spatially Varying-Coefficient Regression Model. In this model, the spatial characteristic factor is added to the data, and the regression parameters change with the change of the spatial position of the sample. Through the research and improvement, the Geographically Weighted Regression (GWR) [35] model is obtained. The Geographically Weighted Regression (GWR) model is an extension of the classical linear regression model, which provides ideas and methods for detecting the heterogeneity of spatial relations, so it is a local regression model. But GWR is still a linear model with the same preconditions as OLS. Strictly speaking, GWR is not a pure model, but with the help of this tool, it can be used to analyze the local changes of spatial phenomena and embed the spatial characteristics of the data into the model. The basic model of GWR [36] is given as Equation (2).

$$y_i = \beta_0(u_i, v_i) + \sum_{k=1}^{p} \beta_{ik}(u_i, v_i)x_{ik} + \varepsilon_i \tag{2}$$

where $(u_i, v_i)$ $(i = 1,2, \ldots ,n)$ is the coordinates of the *i*th sample point (such as latitude and longitude of bird strike point), $\beta_k(u_i, v_i)$ $(k = 1,2, \ldots ,p)$ is the *k*th regression parameter of the *i*th sample point, and $\varepsilon_i$ is the random error of the *i*th sample point, obeys the assumption of normal distribution. And Equation (2) can be abbreviated as Equation (3).

$$y_i = \beta_{i0} + \sum_{k=1}^{p} \beta_{ik}x_{ik} + \varepsilon_i \tag{3}$$

If $\beta_{1k} = \beta_{2k} = \cdots = \beta_{nk}$, the GWR model is the classic OLS model.

The Weighted Least Square (WLS) method is used to estimate the parameters of the GWR model, that is, it is assumed that there are differences in the importance of different sampling observations to point *i* in the field of sampling point *i*. And the difference is measured by the distance between the other sampling points and the point *i*, which means that the point closer to the point *i* is of greater importance, and the farther away from the point *i* is less important. This method minimizes Equation (4) to estimate the regression parameters of the point *i*.

$$\sum_{i=1}^{n}\left[y_i - \beta_{i0} - \sum_{k=1}^{p}\beta_{ik}x_{ik}\right]^2 w_{ij} \tag{4}$$

where $w_{ij}$ is a function of the geographical distance $d_{ij}$ between the sampling point $i$ and the neighborhood observation point $j$, and it is a monotone decreasing function, which is called the spatial weight function.

The spatial weight function calculates the weight of the sampling points by using the distance between the sampling points, which represents the spatial importance of the data. It plays a key role in the parameter estimation of the GWR model. In this paper, the Gaussian function is selected as the spatial weight function, as shown in Equation (5).

$$W_{ij} = \exp(-(d_{ij}/b)^2) \tag{5}$$

where $b$ is the bandwidth, meaning the distance to the neighborhood point $j$, and the reasonable choice of bandwidth $b$ is an essential factor affecting the running results of the model. If the bandwidth $b$ is too large, it will increase the deviation of the estimated value of the regression parameter, and if $b$ is too small, it will increase the variance of the estimated value of the regression parameter. In this paper, the Gaussian spatial weight function with adaptive bandwidth is selected, which is more suitable for the areas where the sample points are distributed unevenly.

However, the bandwidth value needs to meet certain criteria to become the optimal bandwidth, and the generalized cross-validation (CV) method is selected to determine the optimal bandwidth in the paper. The objective function of this method is Equation (6).

$$CV = (1/n)\sum_{i=1}^{n}(y_i - \hat{y}_{\neq i}(b))^2 \tag{6}$$

where $\hat{y}_{\neq i}(b)$ is the fitting value in which the sample point $i$ is eliminated in the regression, and only the nearby sample point is used for regression. When the *CV* value is minimum, the corresponding $b$ value is obtained, which is the optimal bandwidth. Finally, the analysis of the GWR model and the calculation of Moran's I index mentioned above is realized by ArcGIS.

## 3. Results and Discussion

### 3.1. The Spatial Features of Season-Level Bird Strike Estimation

The kernel density estimation (KDE) of bird strike points in different seasons is shown in Figure 2. The high values of four seasons (greater than 0.0447 of the average seasonal level) are all distributed at both ends of the runway, and the proportion of the regions that are higher than the average seasonal level in each season is 23.2%, 23.8%, 39.1%, 7.5%, respectively. The high-value region is divided into two diffused fans at both ends of the runway, and the fan scattered patterns are different in different seasons. The maximum value of KDE varies from season to season. The high-value region is mainly concentrated on the airport and the left of the ends of the runways in spring. And the high-value part is clustered at both ends of the runways and runways in summer, in which the area on the right of runways is larger than that on the left. In autumn, the fan-shaped high-value area becomes more substantial, mainly at the ends of the runways. And the area with high-value is drastically reduced, gathered primarily on the runways. In contrast, low-value areas are aggregated at both ends of the rectangle that far from the runways, and such a spatial distribution is also closely related to the routes of flights. More than 90% of bird strikes occur in the climbing and approach stages [37], in which aircraft are mostly flying at a low altitude below 600 m, so they are most likely to collide with birds in the airport clearance area.

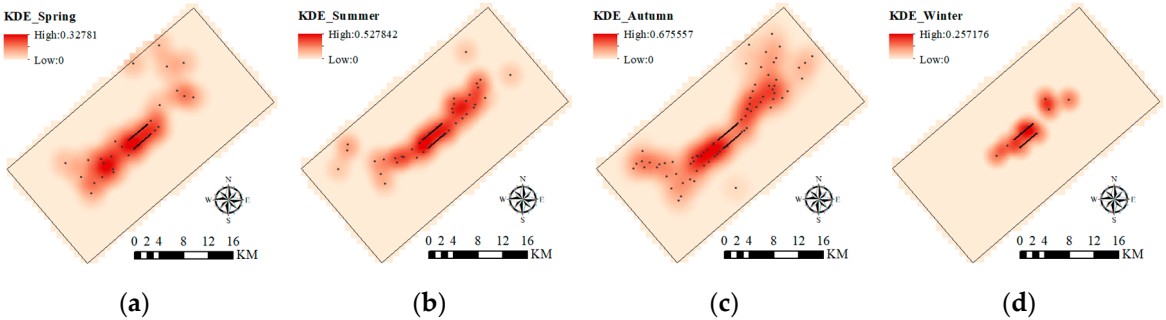

**(a)**          **(b)**          **(c)**          **(d)**

**Figure 2.** Kernel density estimation of bird strikes each season in the study area. (**a**) Spring; (**b**) Summer; (**c**) Autumn; (**d**) Winter.

The global Moran's I index for each season was calculated, as shown in Table 2. The results showed that the Moran's I index are all greater than 0, indicating that there is a positive spatial autocorrelation in the four seasons in the study area. Among them, the spatial autocorrelation is the most obvious in spring, followed by autumn. It also showed that the p-values of KDE are less than 0.01, and the z-scores were less than −2.58 or greater than 2.58, indicating that the confidence degrees of these datasets are 99% and the spatial distribution patterns were significant. In other words, the spatial distribution of KDE in each season were not random, and the KDE with similar characteristics in the study area tended to cluster in space.

**Table 2.** Global Moran's I index of kernel density estimation in different seasons.

| Season | Global Moran's I | z-Score | *p*-Value |
|--------|------------------|---------|-----------|
| Spring | 0.941 | 38.603 | 0 |
| Summer | 0.924 | 37.915 | 0 |
| Autumn | 0.939 | 38.493 | 0 |
| Winter | 0.879 | 36.297 | 0 |

The LISA (local indicators of spatial association) map of KDE shown in Figure 3 describes the local spatial correlation of kernel density geographically in each season. HH clustering regions in all seasons gathered in the center of the studied area and the ends of the runway, areas with larger KDE of bird strikes, while the LL clusters appeared only in autumn during the study period.

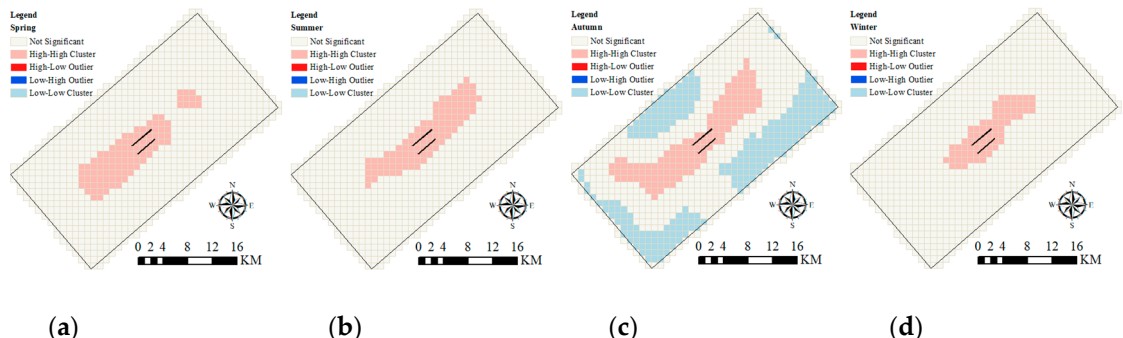

**(a)**          **(b)**          **(c)**          **(d)**

**Figure 3.** The LISA (local indicators of spatial association) maps of kernel density estimation about bird strikes in different season. (**a**) Spring; (**b**) Summer; (**c**) Autumn; (**d**) Winter.

### 3.2. The GWR Model Estimation

We first used the OLS model to observe the global effects of 4 explanatory variables on bird strike, as shown in Table 3. As a result, we found that the VIF (variance inflation factor) values of all the variables were far below 10 [38], and the model has no multicollinearity. The results are

also summarized in Table 3. The coefficients represented that most of the variables have greater or lesser effects on the KDE of bird strikes in different seasons. In particular, the increase in vegetation coverage will cause a significant increase of KDE in summer, specifically, for every 1% incremental increase in the NDVI, the average level of KDE value would increase by 0.26%. On the contrary, the bird diversity index with every 1% incremental increase, the dependent variable would decrease by 0.48%. Although the overall level of the OLS model was significant ($p < 0.01$) in each season, the $R^2$ (goodness-of-fit) of the OLS model in each season was 0.10, 0.22, 0.03, and 0.24 respectively, with a low level of goodness-of-fit and unconvincing.

**Table 3.** Statistics of the OLS (Ordinary Least Square) model in different seasons.

| Variable | Spring | | | Summer | | |
|---|---|---|---|---|---|---|
| | Coefficient | Pr. (> \|t\|) | VIF | Coefficient | Pr. (> \|t\|) | VIF |
| Intercept | -0.000 | 1.000 | | −0.000 | 1.000 | |
| P | 0.059 | 0.143 | 1.57 | −0.026 | 0.471 | 1.43 |
| NDVI | −0.106 * | 0.015 | 1.86 | 0.264 *** | 0.000 | 1.32 |
| BD | −0.321 *** | 0.000 | 1.26 | −0.481 *** | 0.000 | 1.22 |
| RNE | −0.006 | 0.842 | 1.03 | 0.017 | 0.558 | 1.01 |
| $R^2$ | | 0.10 | | | 0.22 | |
| Variable | Autumn | | | Winter | | |
| | Coefficient | Pr. (> \|t\|) | VIF | Coefficient | Pr. (> \|t\|) | VIF |
| Intercept | 0.000 | 1.000 | | −0.000 | 1.000 | |
| P | 0.056 | 0.141 | 1.29 | 0.083 * | 0.014 | 1.33 |
| NDVI | 0.175 *** | 0.000 | 1.29 | −0.036 | 0.284 | 1.34 |
| BD | 0.045 | 0.199 | 1.10 | −0.483 *** | 0.000 | 1.01 |
| RNE | −0.031 | 0.349 | 1.01 | 0.033 | 0.268 | 1.01 |
| $R^2$ | | 0.03 | | | 0.24 | |

Notes: Pr. (> |t|) means p-value; VIF means variance inflation factor of variable; '***' Means $p < 0.001$; '*' Means $p < 0.05$.

As mentioned above in this paper, the OLS model is a global model, which can only represent the effect of explanatory variables on dependent variables in the whole region but cannot explain the spatial heterogeneity and effect direction of independent variables on bird strike kernel density in different geographical locations. Therefore, to determine whether the GWR model is better than the OLS model in this problem, we constructed a GWR model using the same data and select three groups of statistical parameters of the two models to compare. The results are shown in Table 4. The $R^2$ (Goodness-of-fit index of the model) of the GWR model is much higher than those of the OLS model in each season, indicating that the GWR model is more sensitive to the collected data and more suitable for this data. Besides, the RSS (Residual sum of squares) and AIC (Akaike information criterion) of the GWR model in each season are extremely lower than those of the OLS model, so it can be seen that the GWR model shows a better fitting effect than the OLS model in the issue of bird strikes [39].

Another advantage of the GWR model is that it can calculate the regression coefficient of each variable in each region, to depict the spatial variation of the coefficient. The descriptive statistics of the regression coefficients of the GWR model in different seasons are shown in Table 5. The results indicate that the large range of regression coefficients suggests that there is greater spatial variation in the effect of variables in different geographic locations, indicating that GWR models are more suitable for explaining spatial heterogeneity. Furthermore, the standard deviation of the regression coefficient also spans a large range, with the minimum (NDVI in summer) of 0.138 and the maximum (RNE in spring, RNE means the reciprocal of the nearest Euclidean distance between the center of the grid and the water area) of 1.598. The *t*-test ($p < 0.1$) was used to test the significance of the regression coefficient, as shown in Table 5. It can be seen all *p*-value are significant at 90% level, indicating that these factors had varying degrees of influence on the kernel density estimation of bird strikes. Moreover, we also

have tested the significance of regression coefficient at 95% level ($p < 0.05$), the result showed that the $p$-value of the variables in other seasons is less than 0.05, except for the RNE variable in summer. This change indicates that the summer RNE variable is not significant at 95% confidence level, meaning that the water distribution is not an essential factor affecting KDE in most areas in summer of the study year.

**Table 4.** Comparison of fitting parameters between the OLS model and the GWR (Geographically Weighted Regression) model.

| Quarter | Parameter Name | OLS | GWR | Delta Variation |
|---|---|---|---|---|
| Spring | R2 | 0.10 | 0.54 | 0.44 ↑ |
|  | RSS | 806.202 | 407.462 | 398.74 ↓ |
|  | AIC | 2453.177 | 1877.120 | 576.06 ↓ |
| Summer | R2 | 0.22 | 0.45 | 0.23 ↑ |
|  | RSS | 695.992 | 495.773 | 200.22 ↓ |
|  | AIC | 2322.056 | 2052.568 | 269.49 ↓ |
| Autumn | R2 | 0.03 | 0.47 | 0.44 ↑ |
|  | RSS | 867.866 | 473.325 | 394.54 ↓ |
|  | AIC | 2518.920 | 2011.568 | 507.35 ↓ |
| Winter | R2 | 0.24 | 0.50 | 0.26 ↑ |
|  | RSS | 676.372 | 442.595 | 233.78 ↓ |
|  | AIC | 2296.549 | 1949.597 | 346.95 ↓ |

Notes: RSS means residual sum of squares; AIC means Akaike information criterion; ↑ means the increase of parameters; ↓ means the decline of parameters.

**Table 5.** The statistical description of regression coefficients of the GWR model in different seasons.

| Variable | Spring: GWR Coefficients | | | | Summer: GWR Coefficients | | | |
|---|---|---|---|---|---|---|---|---|
|  | Min | Max | Std. Dev | $p$-Value | Min | Max | Std. Dev | $p$-Value |
| P | −0.859 | 0.390 | 0.195 | 0.000 | −0.773 | 0.738 | 0.325 | 0.000 |
| NDVI | −0.930 | 0.598 | 0.324 | 0.000 | −0.127 | 0.470 | 0.138 | 0.000 |
| BD | −0.855 | 0.918 | 0.356 | 0.000 | −0.784 | −0.098 | 0.168 | 0.000 |
| RNE | −34.318 | 0.427 | 1.598 | 0.0002 | −20.488 | 0.869 | 1.226 | 0.022 |

| Variable | Autumn: GWR coefficients | | | | Winter: GWR coefficients | | | |
|---|---|---|---|---|---|---|---|---|
|  | Min | Max | Std. Dev | $p$-Value | Min | Max | Std. Dev | $p$-Value |
| P | −0.395 | 0.499 | 0.252 | 0.000 | −0.455 | 0.753 | 0.279 | 0.000 |
| NDVI | −0.373 | 0.801 | 0.309 | 0.000 | −0.733 | 0.219 | 0.248 | 0.000 |
| BD | −0.366 | 1.426 | 0.347 | 0.000 | −1.011 | −0.014 | 0.303 | 0.000 |
| RNE | −22.027 | 0.140 | 1.300 | 0.000 | -8.926 | 0.840 | 0.588 | 0.000 |

Last, the standard residual histogram of bird strike kernel density in different seasons can be obtained (Figure 4), after analyzing the factors of bird strike kernel density each season by the GWR model. The standard residual is in line with the normal distribution, and is similar to the standard normal distribution. Moreover, we have carried out the KS (Kolmogorov Smirnov) test of the standard residual, and the $p$-value are all greater than 0.05, indicating that the standard residual obeys the standard normal distribution and about 95% of the standard residual is between −2 and 2.

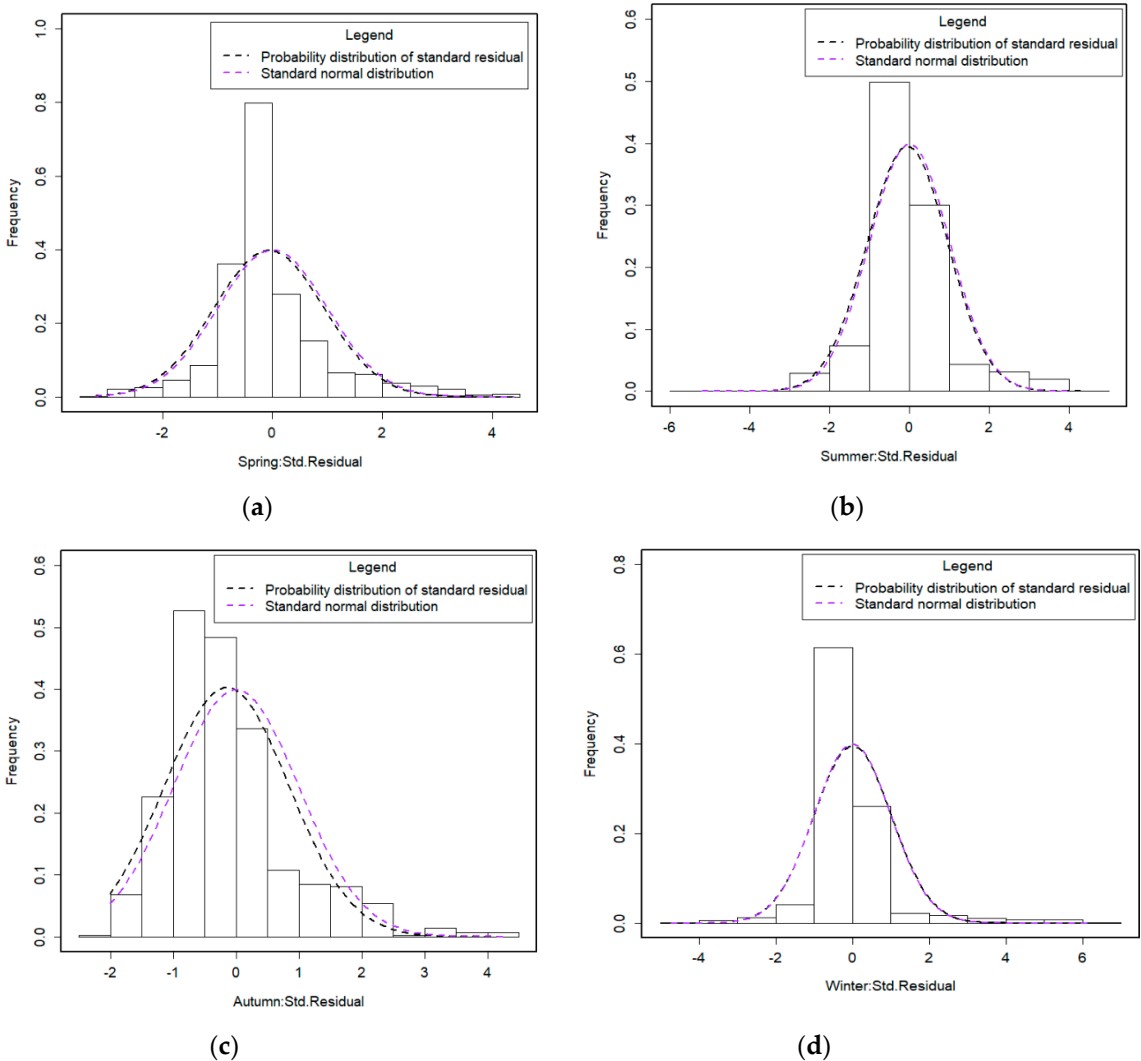

**Figure 4.** Standard residual histogram and probability distribution in different seasons. (**a**) Spring; (**b**) Summer; (**c**) Autumn; (**d**) Winter.

### 3.3. Discussion

The spatial relationships that these influencing factors and the KDE of bird strikes varied differently over space, which cannot be obtained by the OLS model. To further investigate the heterogeneity of the spatial relationship between each variable and the KDE of bird strikes, we depicted the spatial distribution map of the regression coefficients of each variable by season in the following discussion.

As shown in Figure 5, particularly summer and winter, the region with the most significant positive effects (coefficient > 0.4) of the human population density was distributed in the upper left area parallel to the runway, which is partial urban land area. Compared with rural environments, urban avian communities have typically reduced species richness. In contrast, the density of a few successful species is often higher in cities than natural habitats in adjacent [26,40]. Meantime, the closer to the runways areas one gets, the smaller the effect would be, and the regions where had the least effect is the southwest clearance area at ends of the runway. On the contrary, in autumn, the area of positive effects in the lower-left area parallel to the runway, and the degree of effect, is reduced.

Secondly, although the population density involved in the GWR model calculation for each season did not change, the significant area of population density also changed with the change of seasons. The reason for this change owes to the spatial autocorrelation of geographic data and Tobler's first law of geography [32]. Then, these spatial changes of regression coefficients represent the variations of the

relationship between the KDE and human population density at Xi'an Airport. The reason for the more significant positive effect may be closely related to human activities around the airport. For example, modern solid waste produced by human activities is also attractive to many birds, particularly by nuisance birds, which increases the risk of bird-aircraft collision. Based on this, the Federal Aviation Administration of the United States recommends that solid waste management facilities not be located within 8 km of an airport [41]. Moreover, the NDVI was negatively correlated with the human population density but positively related to bird species richness in Taiwan. But there was no evidence that birds and humans responded similarly to productivity [10]. And near the airport was studied, the most common bird species in residential areas were *Passer montanus*, *Sturnus cineraceus*, *Hirundo rustica*. Therefore, the influence of population density on bird strike density in spatial latitude is multifaceted and uncertain. It is necessary to explore the spatial correlation between population density where the certain airport and surrounding landscape and bird strikes, to reveal the complexity and spatial heterogeneity of human effects on bird strike issues at a specific airport.

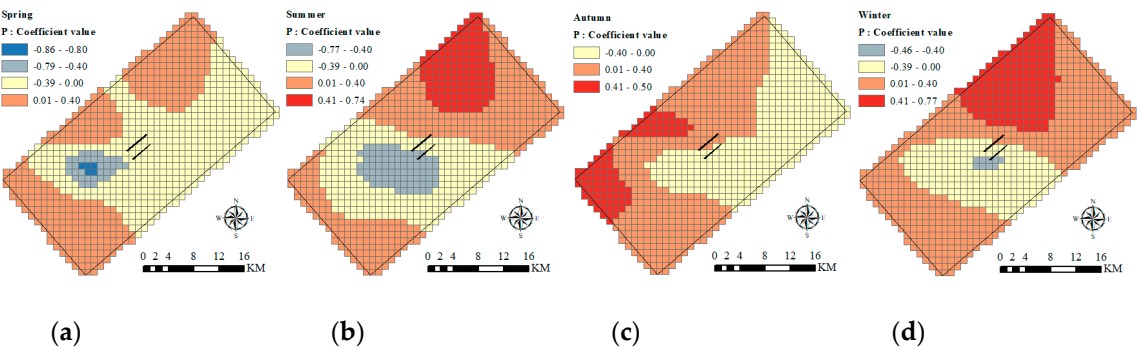

**Figure 5.** The spatial distribution of regression coefficients of the human population density (P) in different seasons. (**a**) Spring; (**b**) Summer; (**c**) Autumn; (**d**) Winter.

As shown in Figure 6, it is found that the effects of normalized difference vegetation index on the kernel density estimation have a significant difference over space. In spring, the significant positive relationship (coefficient > 0.5) between NDVI and bird strike density appeared mainly in the lower right area parallel to the runway, accounting for 1.2% of the study region, and 0, 13.2%, and 0 in other seasons in turn. Notably, in autumn, not only does normalized difference vegetation index had the largest significantly positive (coefficient > 0.5) area of impact throughout all the year but also had the greatest degree of influence. Apparently, the impacts of this factor degraded in regions as they got closer to the runway in each season. In fact, NDVI declined towards the more urbanized areas, for example, Lucas et al. [42], revealed in 2018 that bird richness increased at medium levels of NDVI and was negatively correlated with the seasonal variations of the NDVI. Besides, the NDVI was found to be the most crucial predictor of bird species richness as early as 2006 [10]. As is displayed by the spatial distribution of the GWR local coefficients, the strongest influence of vegetation coverage was mainly clustered in the urban land and nearby.

Obviously, the impacts of bird diversity on bird strike kernel density aggrandized as they got farther from the runway, and the greatest significant relationship between bird diversity and bird density estimation appeared in autumn (Figure 7). In light of our previous analysis, the highest bird diversity of Xi'an Airport also occurred in autumn, indicating that the bird diversity contributed significantly to the variations of the kernel density estimation. Because the surrounding habitats of the airport are closely related to the number and species of birds. Moreover, the dryland accounts for the largest proportion in the area where the regression coefficient is greater than 0.5, indicating that the strongest positive effects of bird diversity clustered mainly in the dryland area. Some other literature found that the bird community structure is different at and around disparate airports, so the bird diversity and its effects on bird strikes are also different. According to statistics, the number of

bird species recorded around Xi'an Airport (excluding the airfield area) in different seasons is 67 in spring, 48 in summer, 102 in autumn, and 41 in winter, respectively. Moreover, Zhalantun Genghis Khan Airport [43] found that the highest bird diversity index appeared in spring within the airport boundary in 2015, while the highest bird diversity index of the region outside the boundary was founded to be in wetland area in summer. However, at the agricultural lands near Kazan International Airport [11], the highest bird diversity index appeared in the onset of migrations (July–September) during the studied period.

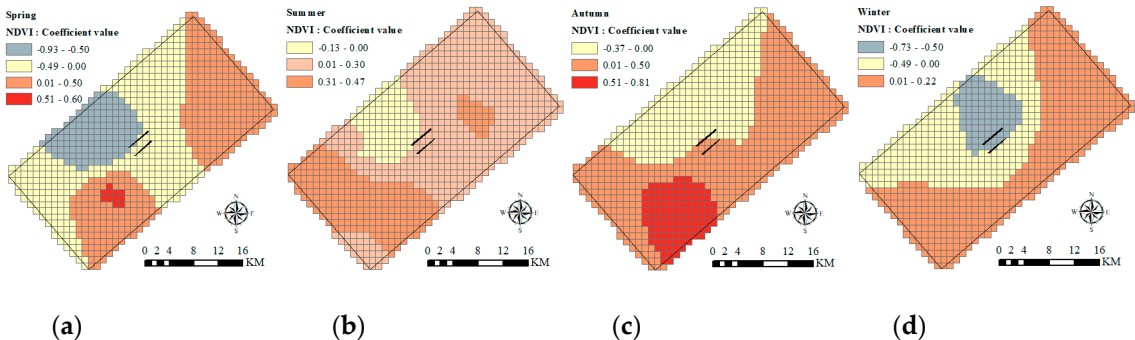

**Figure 6.** The spatial distribution of regression coefficients of the Normalized Difference Vegetation Index (NDVI) in different seasons. (**a**) Spring; (**b**) Summer; (**c**) Autumn; (**d**) Winter.

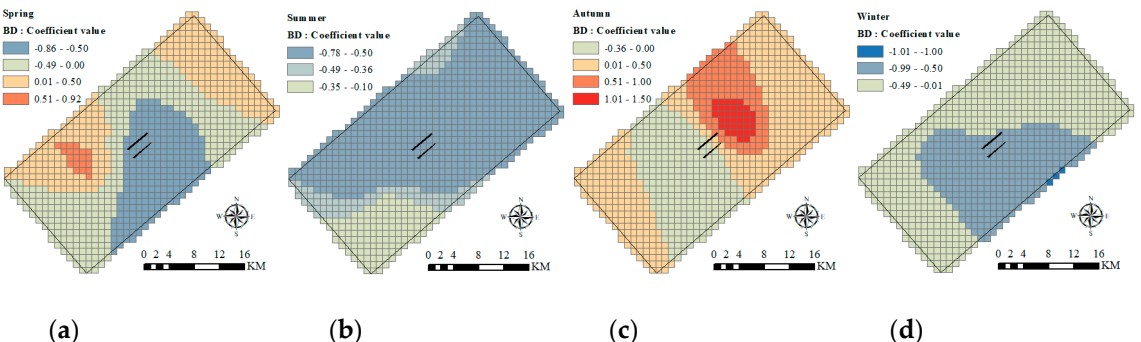

**Figure 7.** The spatial distribution of regression coefficients of the bird diversity index (BD) in different seasons. (**a**) Spring; (**b**) Summer; (**c**) Autumn; (**d**) Winter.

Different from other variables, the spatial distribution of the RNE coefficients (Figure 8) reflected a positive relationship between RNE and bird strike kernel density at the central region studied, among which the region with the positive effects almost completely covered the airport. As revealed by the result from the GWR, the most significant effect of water and wetland was in summer and winter and followed by in spring, and their greatest impacts simultaneously clustered in the northeast part of the airport. According to statistics, 53 species of birds inhabiting waters have been recorded around Xi'an Airport, accounting for 36.1% of the total. Based on the habitat requirements for many species that are harmful to aviation, the aquatic habitat is strongly attracted to a number of waterbirds, such as *Egretta garzetta*, *V.cinereus*, and *Ardea cinerea rectirostris* were found around Xi'an airport. Some similar findings also were found, such as Pfeiffer et al. [15] suggested that managers should increase the distances between patches of open water to reduce the adverse effect bird strike, and Blackwell et al. [44] found that the possibility of birds using the pond outside the distance of 8 km was close to zero.

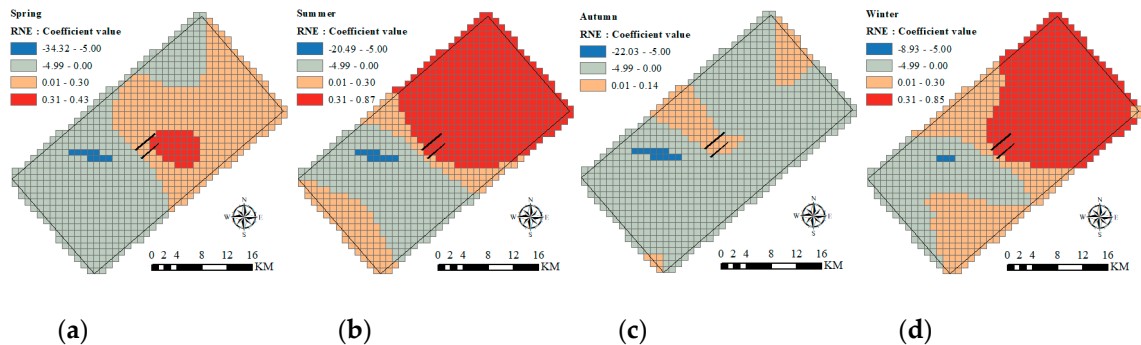

**Figure 8.** The spatial distribution of regression coefficients of the RNE in different seasons. (**a**) Spring; (**b**) Summer; (**c**) Autumn; (**d**) Winter.

As revealed by the result of the GWR, all four factors were found to be seasonal and significant spatial differences in affecting the kernel density of bird strikes. Although the OLS model used in this study also revealed some significant factors in different seasons, its goodness-of-fit and model evaluation factors were much lower than those of the GWR model, based on previous analysis. According to our results from the GWR model (the detailed statistics of all regression coefficients is shown in Table S3), all of the four factors studied showed bidirectional effects that varied between different habitats in most seasons.

At last, according to the above discussion, we can provide some specific preventive measures for the airport management staff, to reduce the attraction of birds and the risk of bird strikes. In terms of spatial region, airport staff usually mainly drove away birds in the airfield area. Based on the existing measures of bird repellent, we suggest to increase reasonably the frequency and coverage of the bird-repellent equipment in the regions with higher regression coefficients to optimize the bird-repellent rate [45,46]. On the other hand, we suggest to reduce the attraction to birds in the habitats outside the airport. For example, it is recommended that airport staff and local managers should jointly supervise and inspect the sowing and drying of farmland areas where the regression coefficients of factors discussed above are larger, such as specific work is to minimize the area of crops or drying of grain [15]. Besides, the aquatic plant and weeds at the junction of rivers should be treated and cleared timely, especially in summer and winter [47,48]. Lastly, the influencing factors that should be paid attention to in different seasons are also various. In the study year, the results showed that the key factors affecting the KDE in the study area were bird diversity in spring and autumn, the water's attraction in summer and winter. For bird diversity, in the nesting and migration period, the population density and diversity of birds are the highest, which will pose a great threat to aircraft. Therefore, it is necessary to regularly take large-scale repelling and frightening measures for birds in spring and autumn [11]. For water distribution, in the non-breeding season of birds, the area of open water, the distance between waters, and other water's shape are important determinants of the existence of water birds, especially geese and gulls [48].

We used data on bird strike, and bird species distribution of the study area from Xi'an Airport as we aimed to explore the seasonal and spatial difference of some factors on bird strike. However, where possible, it is preferable to collect bird data especially for the study (i.e., the data of bird appeared are recorded per habitat categories in a buffer of a rectangle from the airport using a standardized protocol [49]) and to follow an objective sampling scheme, as it improves the accuracy of results. And considering the impact of bird diversity on bird strikes in this paper, we mainly considered the impact of the number and species of birds, not taking into account differences in the escape behavior of species. In subsequent research, we will continue to improve the method to fill the gap in the spatial analysis of unsafe incidents.

## 4. Conclusions

The influencing factors of bird strike events have some spatial differences, but few previous studies focused on the spatial distribution characteristics of these factors. In this paper, the spatial heterogeneities of key factors of bird strike kernel density in different seasons were investigated using Geographically Weighted Regression (GWR) model. The influencing factors, including population density, vegetation coverage, bird diversity, and water distribution in a rectangular region of 800 square kilometers centered on Xi'an Airport runway were considered.

We found that factors in the kernel density of bird strikes showed obvious spatial heterogeneity and seasonal difference. Spatially, in the Xi'an Airport, the bird diversity index was the most significant positive factor in the kernel density of bird strikes, followed by the water distribution. Relatively speaking, the population density of partial urban land was found to exert significant positive effects on the bird strike density. And the strongest positive effects of bird diversity clustered mainly in the dryland area, the strongest positive effects of vegetation coverage agglomerated in the partial urban land away from the airport, but it also had a higher positive effect at the airport and nearby. The water distribution was also found to constitute a significant positive factor, in particular in the northeast part of the airport. For seasons, the key factors that are focused on are the water's attraction in summer and winter, the bird diversity in spring and autumn.

We suggest that future research should focus on continuously improving the research method and adopt our approach on different case study airports (including high-altitude airports in China), in order to explore and compare the spatial effects of environmental and biological factors to varying elevations on bird strikes. We are standing from the perspective of airport habitat to protect the sustainable development of civil aviation security.

**Supplementary Materials:** The following are available online at http://www.mdpi.com/2071-1050/12/18/7235/s1, File S1. The spatial distribution map of each variable for each season in the study area (GIS). File S2. The spatial distribution map (including the number of birds) of main bird species in the study area all year (GIS and XLSX). File S3. Partial approach and departure procedures in this airport (PDF). Figure S1. The legend of the relationship between z-score and *p*-value about Moran's I index (TIF). Table S1. The detailed data of all variables each season (XLSX). Table S2. The *p*-value and z-scores for different confidence levels (XLSX). Table S3. The regression coefficients of variables from the GWR model in different seasons (XLSX).

**Author Contributions:** Conceptualization, Q.S. and Y.Z.; Methodology, Q.S.; Software, Y.Z.; Validation, Q.S. and Y.Z.; Formal Analysis, Y.Z. and P.Z.; Investigation, Y.Z. and M.S.; Resources, Q.S.; Data Curation, Y.M. and M.S.; Writing—Original Draft Preparation, Y.Z.; Writing—Review & Editing, P.Z. and Y.M.; Visualization, Y.Z.; Supervision, Q.S.; Project Administration, Q.S.; Funding Acquisition, Q.S. All authors have read and agreed to the published version of the manuscript.

**Funding:** This research was funded by the National Basic Research Program of China, grant number 2018YFC0809500, the National Natural Science Foundation of China (71874081), and Qing Lan Project.

**Conflicts of Interest:** The authors declare no conflict of interest.

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
