# Peer review of "Key Factors Assessment on Bird Strike Density Distribution in Airport Habitats: Spatial Heterogeneity and Geographically Weighted Regression Model"

_sustainability, doi:10.3390/su12187235_

Round 1

Reviewer 1 Report

I think it is an interesting paper, useful to expand the research focus on bird strikes and to allow a proper selection of useful mitigation measures. However, I think the authors may still include some rationale behind their findings, why are some variables more important during different seasons? I think the paper should also include specific mitigation measures according the results found in the paper. Just a couple of examples were provided by the authors but not references about if those measures have been previously used or tested to decrease bird densities in certain areas. I would expand this section.

As general suggestion, I would try to minimize the use of the passive voice all over the text, and I would suggest to review the English.

Below specific comments and suggestions:

L44:

L47: I would leave the sentence as: A large number of studies have analyzed the influencing factors of bird strikes (REF).

L87: Which kind of professionals?

L92: “and Y-airport is not an exception”

L104: “there are factors that attract birds to the airport environment (e.g. food, water and habitat)”

L106: Which bird strikes, from your study or the ones from references?

L113: “the average population density per square kilometer in a given grid, each season during the study period”

L141: Delete “And”

L195: Please, define bandwidth more specifically. Example: bandwidth is the distance band or number of neighbors used for each local regression equation

L207: Please include which programs do you use for the Moran'sI index and the GWR models. R? ArcGis?

L218-219: Please rephrase.

L232: “in the study area”

L234: 'became smaller by season", is that from spring to autumn? so, you say that differences in the spatial distribution were growing from spring to winter? not sure if I understood correctly your statement. Autumn has a higher Moran's I index than summer, and how the authors know that 0.939 (autumn) is actual different from 0.941 (spring)?

L252: “higher or lower effect”

L252-253: Please rephrase

L254: “increased”

L258: please, include these values on the table, no clear to me to which season are referring to with each value.

Table 3: Are the coefficients the betas of you models?

L268: What do you mean by "significantly" here?

L270: To be able to extrapolate any conclusion from the AIC results you should include your models set and information of the delta AIC, and the AIC weight. The AIC per se does not give any information about a model.

L284: please explain, all your p-values are significant. May you include 95% CIs?

Table 5: min and max indicate the min and max values for each grid?

Figure 4: Please, avoid to use black/red or green/red color combinations for your figures, but instead use a color blind friendly palette.

L305: human population density?

L318: please re-phrase. Do those changes represent changes in the regression coefficients between birds strikes and human population density?

L322: Do the authors have results to support this statement? Would be interesting to test it with your data

L343: Lucas et al., include the year of the publication please

L364: please, indicate which seasons

L401: Please, rephrase

L402: I think here you need cite existing bird-repellent equipment and studies testing them?

L402-403: Rephrase: “we suggest to increase the frequency of the bird-repellent equipment in the regions... On the other hand, in the habitats outside the airport we suggest to reduce the attraction to birds”

L406: May you be more specific here? "Supervise and inspect" but what actions do you suggest to decrease the presence of birds?

L407: Do you know other studies doing this and achieving a reduction of birds’ biodiversity? please include reference

L408: please, rephrase

L409-410: It is that a conclusion only from your results? or from other studies too? please include references. Also, might to give an explanation about why are those factors important each season?

L439: “ We suggest that future research should focus on...”

Reviewer 2 Report

  1. Please enter in the text extended information about the sample (how many collisions with birds were analysed) on which the analysis was performed (describe in the text for particular periods: summer, etc.) .
  2. For Figure 4 the probability density distributions were approximated. The values of statistical tests and the quality of their matching should be indicated.

Very good and interesting paper

Reviewer 3 Report

This manuscript aims to evaluate patterns in bird collisions across different seasons with respect to several suggested drivers. Though it addresses potentially interesting topic, the manuscript is not easy to follow, needs to be better organised, some crucial information is missing and the results are not clearly presented.

Specific comments:
53: local method?
50-51: Try to leave one 'different locations' from the sentence.
41: which -> and?
57: Please specify the 'better result'.
56-58: It is clear that the analysis which takes spatial limitations into account is necessary to study this topic, so you do not need to provide much space to defend its use. Moreover, the provided examples (water quality, urban expansion...) are not much related with the focal topic.
66-69: Instead explaining dependent variables and predictors, please add clearly stated hypotheses supported by published evidence to justify selection of the possible drivers.
81, 324: Avoid the use of "and so on".
77: You must provide as much information as possible about the study site. For me it is not acceptable to hide the identity of the airport - its position and surrounding areas are important for better understanding of any results. For instance, its position with respect to the coast or major migration flyways may play an important role in explaining year-round patterns of bird collisions.
78: Please remove 'with'.
78-81: This belongs to the figure caption.
81: Remove 'it can be seen that'.
83-4, 92-93: Please provide more details (especially the numbers of collisions, see also L 95).
95: Did you take number in birds per collision into account?
101: I suggest to present all seasons in the supplement.
Table 1: You should consider other variables as well - e.g. distance to the closest vegetation structures (e.g. trees or shrubs).
127: Details on quantitative survey completely missing.
210-211: This sentence is redundant.
212: What this means? Please also add absolute numbers of collisions and claims which are easy to follow for those who do not fully understand the statistical background.
216-217: How it differs between seasons?
227-228, 236-238, 248-250: This belongs to methods, please also refer to the tables/figures ideally using a link, not a whole sentence (see also 107-108).
254: Please describe what it means (e.g. talk about vegetation greenness).
Fig. 4: I suggest to move this to the supplement.
301-304: Discussion should start with a brief summary of main results, not with a long reference to the Supplement.
305-306: This belongs to the Figure caption. Why is this information not presented in the results section?
312: West/east will be better than left/right.
316-317: Please add more clear explanation and/or discuss weaknesses of your study design.
321-323: Too vague statement - of course, human activities largely affect the distribution of vegetation.
Fig. 5, 6: The figures should be self-explanatory and here it is not clear what this means, i.e. what the regression coefficients represent. Present probabilities of collisions instead or explain the scale of the reg. coefficients.
Fig. 7: Please explain the scales (what low values mean and vice versa).
342-345: Please split into two sentences.
404: Provide references supporting claims and suggestions within this paragraph. Perhaps some suggestions could be found at www.conservationevidence.com
440: China
